# Fungi as a Source of Edible Proteins and Animal Feed

**DOI:** 10.3390/jof9010073

**Published:** 2023-01-03

**Authors:** Amro A. Amara, Nawal Abd El-Baky

**Affiliations:** Protein Research Department, Genetic Engineering and Biotechnology Research Institute (GEBRI), City of Scientific Research and Technological Applications (SRTA-City), Universities and Research Center District, New Borg El-Arab City P.O. Box 21934, Alexandria, Egypt

**Keywords:** fermented foods, fungi, mushrooms, single cell protein, yeast

## Abstract

It is expected that the world population will reach 9 billion by 2050. Thus, meat, dairy or plant-based protein sources will fail to meet global demand. New solutions must be offered to find innovative and alternative protein sources. As a natural gift, edible wild mushrooms growing in the wet and shadow places and can be picked by hand have been used as a food. From searching mushrooms in the forests and producing single cell proteins (SCP) in small scales to mega production, academia, United Nations Organizations, industries, political makers and others, play significant roles. Fermented traditional foods have also been reinvestigated. For example, kefir, miso, and tempeh, are an excellent source for fungal isolates for protein production. Fungi have unique criteria of consuming various inexpensive wastes as sources of carbon and energy for producing biomass, protein concentrate or amino acids with a minimal requirement of other environmental resources (e.g., light and water). Fungal fermented foods and SCP are consumed either intentionally or unintentionally in our daily meals and have many applications in food and feed industries. This review addresses fungi as an alternative source of edible proteins and animal feed, focusing mainly on SCP, edible mushrooms, fungal fermented foods, and the safety of their consumption.

## 1. Introduction

Fungi include yeasts, rusts, smuts, mildews, molds, mushrooms, and toadstools (harmful mushrooms). They are eukaryotes that comprise approximately 80,000 recognized species. Fungi are among the most widely distributed organisms on earth [1,2]. They are of environmental and medical importance. They contribute to degrading nearly all hydrocarbon wastes. Many fungi are free-living, parasitic or symbiotic with bacteria, plants, or animals. Fungi can be distinguished by their principal modes of vegetative growth and nutrient uptake. Fungi grow from the tips of filaments (hyphae) that make up the bodies of the organisms (mycelia). They digest organic matter externally before absorbing it. Alone or with the collaboration of bacteria, fungi break down organic matter and release carbon, oxygen, nitrogen, and phosphorus into the soil and the atmosphere [3]. Based on their structure and life cycle, they can be classified into five groups: Ascomycetes, Basidiomycetes, Zygomycetes, Oomycetes, and Deuteromycetes [4]. Different fungal species from the genera *Actinomucor*, *Amylomyces*, *Mucor*, *Rhizopus*, *Monascus*, *Neurospora*, *Aspergillus*, *Penicillium*, *Candida*, *Endomyces*, *Hansenula*, *Saccharomyces*, *Torulopsis*, *Trichosporon*, *Zygosaccharomyces* and others are reported to be involved in biotechnological food applications [5].

Food production is based mostly on the agricultural activities. Nevertheless, during the last 60 years, only a 10% increase in agricultural production has been reported [6], which did not reflect the human demand due to population pressures and urbanization [7]. Dietary protein (either plant-based or animal-derived proteins) is essential as it provides amino acids which cannot be synthesized by humans’ or animals’ bodies [8]. The world shortage of animal-derived proteins is a key problem [9]. Additionally, plant-based protein sources, for instance beans, are nutritionally valuable protein sources but will face limitations to meet the global demand for protein as they need arable land and water. As a result, intense continuous efforts have been made since the early fifties via exploration of innovative, alternative and exceptional protein sources. In 2013, Boland et al. studied the growing demand for meat and dairy proteins and the urgent need to improve animal production to match the increasing demand sustainably, along with finding and accepting novel sources of protein, both as animal feed and for direct consumption of humans [9].

The term single cell protein refers to any protein from microbial sources in the form of biomass or extracted protein [10]. SCP are produced with the intention of using them as substitute for protein-rich foods (either plant-based or animal-derived foods) for humans and animals. Various microorganisms and substrates are used to produce SCP. For ages, microorganisms have been used for food production and animal feed supplementation but using them in SCP production is a modern concept [11,12,13,14]. In general, microorganisms are unique by their ability to upgrade low protein content of fermented foods [15]. Various microorganisms are used for the production of single cell proteins; bacteria (e.g., *Rhodobacter capsulatus*), yeast (e.g., *Saccharomyces cerevisiae*, *Pichia pastoris*, *Candida utilis*, *Torulopsis glabrata*, and *Geotrichum candidum*), algae (e.g., *Spirulina* (dietary supplement), and *Chlorella*), and molds (such as *Aspergillus oryzae*, *Fusarium venenatum, Trichoderma*, and *Rhizopus*) [13]. Filamentous fungi are easy to harvest from the SCP fermentation medium and fungi including yeasts can also provide vitamins of the group-B. They have cell walls rich in glucans that add fiber to the diet. However, fungi have their limitations. Their growth rates and protein content are lower relative to other microorganisms, with moderate nucleic acid content that is too high for consumption of humans and needs, additional costly processing steps to decrease it, and not being publicly accepted [13].

Fungi have been used traditionally to produce various fermented foods and beverages [16]. Traditional fermentation processes that involve fungi and yeast include producing soy sauce, miso, tempeh, mold-cheeses and beverages such as beer, wine and spirits. Mushrooms, the fruiting bodies of macrofungi, are also important foods with high nutritional (low in calories and rich in proteins, vitamins, and antioxidants) as well as culinary value [17].

Nowadays, there is a significant number of companies which produce microbial proteins used in the food applications. The number of patents in the microbial protein production reflects the demand. Hüttner et al. (2020) reported that out of 324 identified patents concerning food products, 38% have been owned by the top ten organizations [18]. The key players have been DuPont (47 patents), DSM (16 patents), AB Enzymes (13 patents), Novozymes (11 patents), and Toray Industries (10 patents). Marlow Foods (UK) already has seven meat alternatives patents based on filamentous fungi.

Another approach that supports the global microbial protein production is the mushrooms production. Global mushroom cultivation has been estimated at approximately 11.9 million tons per year in 2019 [19]. China alone produced 8.9 million tons of them, followed by Japan (0.47 million tons) and the USA (0.38 million tons).

Asia patents profoundly focused on traditional fermented products (e.g., Kikkoman and Yasama), Europe and the USA patents focused on the protein shift towards mycoprotein as a complete food source [18]. Beside the SCP, the term mycoprotein takes its place and refers to the protein-rich food made of filamentous fungal biomass that can be consumed as an alternative to meat. Mycoprotein is characterized by its low fat and high protein and fiber content [20,21]. It shows positive effects on the blood cholesterol level [22]. A glycemic response (relating to the effect of different foods on blood sugar levels) has been reported [23].

European food law prominently influences the transformative potential of alternative proteins, including SCP. The Novel Food Regulation could be challenging for small companies, and even for larger ones, as it is considered time-consuming and demanding. The transformative potential of all novel and traditional foods is more diminished from third countries. The genetically modified (GM) Food Regulation is scientifically and procedurally demanding, and it makes GM labeling a must. From the viewpoint of business, the process of health claims is equally challenging as the process of novel foods [24].

Genetic modification of the microorganisms that produce SCP can improve the nutritional value of SCP [25] and alter the tolerance of microorganisms to several growth substrates [26]. Microbes might also be genetically engineered to produce dairy proteins such as whey or casein to substitute traditional dairy products. The best example of dairy substitute created from GM yeast is the animal-free ice cream launched by the company Perfect Day, Inc. (www.perfectdayfoods.com, accessed on 30 June 2021) in the USA in 2019. However, the strict EU GM Food Regulation must be applied to GM microbial proteins.

The Novel Food Regulation is concerned mainly with the nutritional and food safety concerns with foodstuffs for human consumption. In case of SCP, the chief concerns of food safety are toxic metabolites (e.g., mycotoxins), the high content of ribonucleic acid (RNA), besides microbial culture contamination with other microbes [13]. If SCP are produced in the form of extracted proteins, the extraction process may significantly change the nutritional content of the raw materials and the final protein isolate may therefore be considered a novel food, while the microorganisms that produce SCP would not fall under Novel Food Regulation (Regulation (EU) 2015/2283 [27]).

There are three fungal strains (SCP producers) that are accepted for food use in EU countries. The first is *Saccharomyces cerevisiae* (Brewer’s yeast, or budding yeast), which has been consumed in EU countries before 1997. The second is Quorn (mycoprotein of the microfungus *Fusarium venenatum*). In 1985, Quorn was introduced to the market in the UK and was widely distributed in EU countries during the 1990′s [28]. It is possibly the largest brand of meat alternative in the world. Quorn came to the market of EU countries before the Novel Food Regulation [28]. The third fungal strain (the yeast *Yarrowia lipolytica*) has been authorized via the Novel Food Regulation ((EU) 2017/2470), yet its use is restricted to food supplements.

Another fungal SCP product is PEKILO (mycoprotein from *Paecilomyces variotii*), which was used as feed for poultry and fish. It was first developed in the 1960′s to valorize pulp and paper industry side streams [29]. Until 1991, PEKILO was commercially produced and presently, the PEKILO production is possessed by a start-up company eniferBio (https://www.eniferbio.fi/, accessed on 6 May 2021). Though PEKILO was first produced to be used as protein rich feed, its potential use as food ingredient was also studied [29].

The current review will address fungi as an alternative protein source. SCP production from yeasts and filamentous fungi for consumption of humans and animals, edible mushrooms, selected fungal fermented foods, and their safety of consumption concerns will be discussed. Important fungal species are included with the names of their commercial products that are mostly protein (mycoprotein).

## 2. Single Cell Proteins

Application of microorganisms for food production is ancient. Since 2500 BC, yeast (*Saccharomyces* ssp.) has been used in bread and beverage making [30]. Then, methods have been developed to produce high concentration of yeast in 1781. During the first century B.C., edible mushrooms have been extensively consumed in Rome. Germans used *Candida utilis* in soups during first and second world wars. By 1967, *Candida utilis* in soups has been produced on an industrial scale [31]. Torula yeast and brewer’s yeast, a byproduct of the brewing industry, are broadly available as food supplements.

In the 1960s, researchers at British Petroleum established a novel technology named “proteins-from-oil process” that involves production of microbial protein from yeast fed with waxy n-paraffin, a byproduct of oil refineries with a capacity to produce 10,000 tons. By the 1970s, the idea of “food from oil” became rather popular, with the UNESCO Science Prize won by Champagnat in 1976, and a number of countries have built paraffin-grown yeast facilities. This product has been primarily used as poultry and cattle feed. Carol L. Wilson invented the term “SCP” in 1966 to replace the term “microbial protein” [32,33].

Much of the recent interest in SCP is focusing on improvement of the quality of the produced protein. There is also increased interest in utilization of mixed populations, instead of pure strains in the production of SCP. Furthermore, some low-cost substrates used for SCP production is reaching commercial scales and more protein-rich products are being produced for both food and feed [13].

Until now, SCP provides a moderately small proportion of human nutritional needs, but the growing global demand for protein will probably make SCP increasingly essential [9]. Large-scale SCP production has unique features, including: (1) wide variations in involved microorganisms, raw materials, and methodologies; (2) high productivity, fast growth rate, and the ability to utilize unique substrates such as CO_2_ or methane; (3) highly efficient substrate conversion; and (4) independence of seasonal factors [34,35].

The SCP production process involves general steps: (1) nutrient media preparation, mostly from inexpensive wastes as straw, wood, cannery, food processing wastes, fruit and vegetable wastes (food processing leftovers), low quality fruits, hydrocarbons, or residues from alcohol production; (2) cultivation and fermentation; (3) separation and concentration of SCP, drying; and (4) final processing of SCP into products for food/feed applications [13].

SCP are used in fattening of calves, pigs, and poultry, and fish breeding. For human consumption, SCP are used as vitamin and aroma carriers, emulsifying aids and to upgrade the nutritive value of baked products in soups [36]. At the industrial scale, regulatory issues must always be considered.

Nevertheless, SCP development as a major food source is limited by problems including (1) their need to be processed to eliminate bitter or unpleasant tasting materials, (2) their digestibility varies with the microbial source, (3) the high nucleic acid content, (4) toxic materials and pollutants may contaminate SCP, and (5) SCP are generally deficient in the dietary essential amino acids lysine and methionine [36]. Yet, small-scale or household production of some SCP products may become feasible, in much the way that homemade yogurt production or mushroom cultivation have been successfully established everywhere [37].

SCP, especially those intended for use as human food are commonly produced from substrates of food grade. Processes will hopefully be developed to produce SCP from inexpensive wastes from the processing industries of food and beverage in addition to directly from agricultural and forestry sources [38]. Currently, SCP for human consumption are produced from a restricted number of microbial species and food grade substrates. As will be mentioned in this review, products from yeasts and filamentous fungi are currently in use or under development.

### 2.1. SCP Production Systems with Different Substrates and Processes

The application of SCP in animal feed may facilitate their development into products appropriate for human consumption. The EU catalogue of feed includes numerous products of microbial origin (Commission Regulation (EU) No 68/2013). Furthermore, the feed law also addresses the growth substrates accepted for each microbial strain. The used substrate in SCP production greatly affects the nutritional content of SCP products and presence of contaminants in these products. Moreover, the used substrate can determine the SCP carbon footprint and sustainability. From the point of view of feed safety, the regulation of appropriate production processes and substrates for SCP is apprehensive.

Commission Regulation (EU) No 68/2013 contains three crude protein products from fermentation (by-) products from yeast and filamentous fungi, which are approved as feed. The first approved crude protein is produced from *Saccharomyces cerevisiae*, *Saccharomyces ludwigii*, *Saccharomyces carlsbergensis*, *Saccharomyces uvarum*, *Kluyveromyces fragilis*, *Kluyveromyces lactis*, *Candida utilis*/*Pichia jadinii*, *Torulaspora delbrueckii*, or *Brettanomyces* ssp. grown on substrates typically of vegetable origin such as sugar syrup, molasses, alcohol, cereals, distillery residues, and products containing fruit juice, starch, lactic acid, whey, sugar, fibers of hydrolyzed vegetable and fermentation nutrients such as mineral salts or ammonia. The yeast cells have been inactivated or killed. The second crude protein is obtained from the yeast *Yarrowia lipolytica* grown on vegetable oils and degumming and glycerol fractions formed during production of biofuel. The cells of *Yarrowia lipolytica* have also been inactivated or killed. The third crude protein is a fermentation by-product of *Aspergillus niger* (fungal cells should be inactivated or killed) on malt and wheat for enzyme production.

In addition, standardized analysis and identification procedures for SCP are critical to guarantee the food and feed safety. Therefore, the European Food Safety Authority (EFSA) established a technical committee that defines standards and practices for algae and their products. Nevertheless, common standards and practices should be established for SCP from other microbes. Furthermore, in the future, the EU should come to a decision whether the regulation of SCP will focus on the final product itself or regulation will extend to the used growth substrates in the production [24]. In 2014, Enzing et al. reported that the EU regulations on GM microbes are more restrictive compared to American rules [39]. The EU regulations also focus on the use of particular SCP technology, rather than on final food product safety as in the case of the American rules.

The large-scale development of SCP production processes worldwide added greatly to the progress of current biotechnology and provided technical solutions for other related technologies. Many fields have been involved in research and development of SCP production processes including microbiology, genetics, biochemistry, food technology, chemical and process engineering, animal nutrition, agriculture, medicine, toxicology, veterinary science, ecology, and economics.

The production of SCP occurs mostly in a fermentation process, in which designated strains of microorganisms are multiplied on appropriate raw materials (substrates) in technical cultivation process leading to the growth of the biomass followed by separation processes. Suitable production strains are screened. Then, the technical cultivation conditions for the selected strains are optimized and done. All metabolic pathways and cell structures should be determined. In addition, process engineering and apparatus technology adjust the technical process performance so as to make the production ready for use on the large technical scale. Here, economic factors as energy and cost start to come into play. Safety demands and environmental protection with regard to the process and the product are considered in SCP production. Finally, legal aspects as operating licenses, legal protection of new processes and microbial strains, and product authorizations for particular applications are considered [40].

The typical used raw materials are substances containing mono and disaccharides. Nevertheless, these materials have a high price level, which adds to the cost of the production of microbial biomass [41]. Thus, substrates that are normally abundant and those where the carbon and energy source is derived from should be chosen in the design of SCP processes. Companies that produce SCP such as Kanegafuichi (Japan), BP (UK), and Liquichimica (Italy) appeared on the scene [42]. In the USA, about 15% of the SCP-making plants or less depend on hydrocarbons as carbon and energy source. There are many other potential substrates for SCP production including citrus wastes, bagasse, sulphite, whey, molasses, waste liquor, animal manure, sewage, or starch [42].

The use of energy sources of high commercial value such as methane, gas oil, n-alkanes, and methanol is of interest in SCP production. British Petroleum grew two yeasts, *Candida lipolytica* and *Candida tropicalis* on C_12_–C_20_ alkanes as substrates. The produced product has been named TOPRINA and has been tested for toxicity and carcinogenicity for 12 years [43]. This product has been marketed as a substitute for fish meal in high protein feeds and in milk replacers as an alternative for skimmed milk powder. However, Japan has been the first country to prohibit petrochemical SCP. By 1977, Italy also stopped the use of alkanes in SCP production because of high oil prices and the more competitive price of soya. Currently, no factory produces any petrochemical SCP.

Amoco Company in the USA grew food grade yeast; *Torula* on ethanol. This product is sold under the trade name “Torutein” in the USA, Sweden, and Canada and contains about 52% protein. It is marketed as a flavor enhancer and an alternative for meat, egg, and milk proteins. Nevertheless, in the USA, soya is more abundant and cheaper alternative to meat and egg diets [14].

#### 2.1.1. SCP from Wastes

The industrial large-scale production of SCP is greatly affected economically by the used carbon substrate. If the used carbon substrate is locally available waste substrate, no general consideration of cost-effectiveness is possible. However, if it is a synthetic raw material as in the case of glucose (hydrolyzed starch), methanol, and ethanol, etc., calculation can be more accurate. SCP production needs accurate evaluation of costs because they compete against plant-based protein sources, which are predictably cheaper. The costs of the used substrate are the largest single cost factor (more than 60% of the total product cost). In addition, the variation in the substrate costs affects the total manufacturing costs. Thus, simple manufacture and purification of raw material, especially in larger plants, can save costs. There are four factors involved in the costs of raw materials, site, capacity of the plant, substrate yield, and product concentration [13,36].

SCP production can contribute to getting rid of a very high amount of local agricultural and some industrial wastes besides creation of edible protein. Cellulose derived from agriculture and forestry sources represents the most plentiful renewable resource on earth as potential substrate for SCP production. However, it is usually found in a complex form in nature either with lignin, starch, or hemicellulose. Consequently, it must be pretreated either chemically (acid hydrolysis) or enzymatically (cellulases) to be used as a substrate for SCP production [44,45].

Several studies have reported SCP production from filamentous fungi and yeast grown on lignocellulosic substrates (e.g., sugarcane bagasse, cactus pear, lignocellulosic hydrolysates, and others) [46,47,48,49]. In 2016, Somayeh et al. produced single cell protein from *Saccharomyces cerevisiae* (PTCC 2486) grown on sugarcane bagasse [47]. They obtained 0.078 g protein/g substrate from the fermentation process. Wu et al. (2018) produced single cell protein from a novel yeast strain, *Candida intermedia* FL023, from lignocellulosic hydrolysates [49]. They suggested feasibility of feed and food additive production from the abundant lignocellulosic bioresources.

Spent sulfite liquor derived from cooked wood in a medium containing calcium sulfite has been used as a fermentation substrate in Sweden since 1909 and in many other countries later. Firstly, the involved microbe has been *Saccharomyces cerevisiae* even with its inability to metabolize pentoses, which are plentiful in this waste product. Later, it has been replaced with *Candida tropicalis* and *Candida utilis,* which have this missed ability. In 1983, the produced biomass from this process has been estimated to be about 7000 tons per year [41]. In 2010, Pereira et al. produced single cell protein by *Paecilomyces variotii* [50]. *P. variotii* was cultivated directly in spent sulphite liquor achieving a high biomass concentration. They quantified yield of biomass and protein content and analyzed nucleic acid concentration to confirm the possibility of fungus biomass being approved as single cell protein for commercial use. The achieved yield of fungus biomass was 1.26 mg biomass/mg carbon consumed and protein yield was 0.41 mg protein/mg biomass. The produced single cell protein contains 41.3 ± 1% protein and only 12.8 ± 4% of nucleic acids. They suggested that the biomass of *P. variotii* can be sold as SCP, an added-value product for animal nutrition.

*Chaetomium cellulolyticum* is a cellulolytic fungus that has a faster growth rate, better amino acid composition, and forms 80% more biomass protein than *Trichoderma*, another high cellulase-producing fungus [51]. This fungus is the most suitable for SCP production from cellulose.

Starch represents a cheaper and more agreeable SCP substrate. This very abundant carbohydrate can be found in maize, rice, and cereals as well as in wastes from root crops, cassava, and potatoes. Starch can be hydrolyzed to fermentable sugars via enzymes from yeast and molds. In Sweden, the Symba process used starchy wastes and two yeasts in sequential mixed culture: *Endomycopsis fibuligira*; amylase producer, and *Candida utilis*; the faster grower [52,53]. Liu et al. (2014) reported conversion of waste of potato starch processing into single cell protein in a two-step fermentation process [54]. The produced single cell protein can be used as high-quality feed. They used a mutant strain of *Aspergillus niger* (*Aspergillus niger* H3) with more cellulase productivity. After treatment with *A. niger* H3, the rate of cellulose degradation of potato residue reached 80.5%. The achieved protein content was 38%. A liquid fermentation using *Bacillus licheniformis* was conducted as the second step.

Wastes from several fruits such as sweet orange, orange, banana, mango, pomegranate, grapes, pineapple, papaya, watermelon, and many others are potential substrates for production of SCP. These SCP products can be used as animal feed or a protein supplement in human food [53]. Mondal et al. (2012) used cucumber and orange peels (food processing leftovers) as substrates to produce single cell protein from *Saccharomyces cerevisiae* in submerged fermentation [55]. Results revealed that fruit leftovers were highly susceptible to hydrolysis. Cucumber peel produced higher amount of protein (53.4%/100 g of substrate) compared to orange peel (30.5%/100 g of substrate). Jaganmohan et al. (2013) reported that *Aspergillus terreus* has a high protein value and can be a good choice for SCP production using Eichornia and banana peel as substrates [56]. Fatmawati et al. (2018) used banana peel as substrate for production of single cell protein from *Saccharomyces cerevisiae* Y1536 and *Rhizopus oryzae* FNCC 6157 to be utilized as fish feed [57].

Whey is commonly derived from the curding process in cheese production. Whey has been considered as a particularly proper substrate for SCP production [58]. In 1956, Fromageries Bel; a French dairy company established a project for yeast production from whey, using *Kluyveromyces marxianus* (a lactose metabolizing yeast) [41,59]. Yadav et al. (2016) characterized and recovered residual soluble proteins after cultivation of monoculture of the yeast *Kluyveromyces marxianus* and mixed culture of *K. marxianus* and *Saccharomyces cerevisiae* on whey, to serve as food-grade single cell protein [60].

Molasses, a byproduct of the sugar manufacturing process is used for biomass production but when supplemented with an appropriate nitrogen source and phosphorus [44]. In the UK, *Fusarium graminearum* grown in molasses or glucose supplemented with NH_3_ as nitrogen source as well as pH control is marketed as pies and is considered a success as it has less fat than meat. Hashem et al. (2022) screened and adapted four non-conventional yeast strains (*Hanseniaspora uvarum* JQ690236, *Hanseniaspora guilliermondii* JQ690237, *Cyberlindnera fabianii* JQ690242, and *Issatchenkia orientalis* JQ690240) to produce SCP at high productivities and yields from molasses of wasted date [61]. They reported that these newly isolated yeasts are promising SCP producers for possible use as animal feed.

Coffee-pressing wastes contain soluble carbohydrates and have a high chemical oxygen demand (COD) and soluble solid contents. In Guatemala, *Trichoderma ssp*. is used for production of SCP on this substrate [52]. Examples of different fungal strains that utilize different types of waste as substrate for SCP production are presented in Table 1.

#### 2.1.2. Fermentation Process

The fermentation process involves certain requirements: (1) the chosen organism in a pure culture in the correct physiological state (in case of submerged fermentation); (2) sterilized growth medium (in case of submerged fermentation); (3) a production fermenter equipped with an aerator, a stirrer, thermostat, a pH detector, and such; (4) cell separation; (5) collection of cell-free supernatant; and (6) product purification and effluent treatment [90,91].

For the production and harvesting of SCP, cost is a major problem. This is because even high-rate production results in diluted solutions (≤5% solids). Thus, concentrating the solutions should be performed via precipitation, filtration, use of semi-permeable membranes, or centrifugation. These processes are expensive and will not be suitable for small-scale production. SCP should be dried to 10% moisture or alternatively condensed and denatured to avoid spoilage [91].

The physiological characteristics of microorganisms suggest control of concentration of carbon sources as a limiting substrate, in addition to an acceptable supply of oxygen to sustain balanced growth under an oxidative metabolic pattern. Considering the fact that microbial growth is a time-dependent process, a suitable technology that maintains proper growth conditions for a prolonged period of time must be implemented to achieve high yield and productivity. Therefore, batch fermentation is not suitable for biomass production, as the conditions in the reaction medium change with time [41].

Fed-batch fermentation is superior for biomass production because it involves carbon source supply control via feeding rates. Nevertheless, with the increase in biomass concentration, the culture oxygen demand reaches a level that cannot be met in engineering or economic standing. Fed-batch culture is used in *Saccharomyces cerevisiae* production [40] but is not preferred for SCP production at a large industrial scale.

The chemostat principle, the continuous addition of fresh medium with the simultaneous product harvesting, has been implemented effectively in industrial fermentation for biomass production [92]. Using this fermentation, production periods of many fungi and yeast reached six weeks [93]. Continuous operations were found to be the most cost-effective ones, thus most processes of SCP implemented on an industrial scale are adjusted to continuous design. For more details, refer to Ritala et al. (2017) and the references within [13]. As an example, Quorn was produced by batch cultivation of *Fusarium venenatum* strain A3/5 on starch and other wastes in 1964. Afterwards, its production process was subjected to an evolution of 20 years and an estimated expenditure of R&D of USD 0 million. In 1985, the UK Ministry of Agriculture, Fisheries and Foods granted Quorn unrestricted use. It is currently produced in continuous culture and the biomass is handled to achieve similar taste and texture to those of meat products.

Ther appearance of foam on the head space of the reactor is a common problem encountered in industrial fermentation, resulting in spillages, reactor pressurization, and contamination threat. Thus, the air-lift fermenter and the deep-jet fermenter have been introduced [94]. The air-lift fermenter achieved great success in continuous SCP production. This is still in use for mycoprotein production in Quorn products.

The crucial process variables must be controlled in SCP production, starting from substrate concentration, oxygen transfer, and product concentration to the appearance of minimal amounts of toxic compounds from undesired metabolic processes to adjust quality of the final product.

The yeast biomass is typically harvested by continuous centrifugation, while filamentous fungi by filtration [95]. The biomass is then treated to reduce RNA and dried in steam drums of spray driers.

In solid state fermentation (SSF), microorganisms are grown on insoluble substrate (wastes) where there is no free liquid. This favors fungal growth. Fungal growth in SSF achieves a much higher biomass concentration and protein yield when compared to submerged fermentation. The SSF process skips elaborate prearrangements for media preparation. Currently, there is a worldwide interest for SCP production using SSF [47,56,57,63,64,67,96]. There is an active and ongoing research and development on SCP from various fungal species, which may result in innovative products or production processes. Much of the current research focuses on the use of various waste substrates such as common food industry wastes for production of fungal SCP [67,74,75]. Aggelopoulos et al. used SSF instead of submerged cultivation for production of fungal SCP as animal feed [67]. Muniz et al. (2020) used solid state fermentation to produce SCP from *Saccharomyces cerevisiae* cultivated on guava peels and cashew bagasse and then produced SCP was included on three different formulations of cereal bars for human nutrition [96]. The SSF of fruit byproducts was performed with 70% equilibrium humidity at 30 °C, 0.9 water activity, and initial concentration of yeast of 3% in case of cashew bagasse and 5% in case of guava peels. SSF increased protein content of both byproducts by 11 times. All cereal bars supplemented with produced SCP showed average scores of 7/10 for sensorial attributes and 4/5 for purchase intention. They suggested that the addition of SCP is an alternative to adding nutritional and economic value to cereal bars.

### 2.2. SCP from Yeasts

Vegemite is an Australian yeast extract flavored with vegetables and spices made from Spent brewer’s yeast (*Saccharomyces cerevisiae*), which has been sold since 1922. Vegemite is currently manufactured by Bega Cheese (Bega, New South Wales, Australia). Marmite (a British yeast extract produced by Unilever company, London, UK), *VITAM*-*R* (yeast extract produced by VITAM Hefe-Produkt GmbH company, Hameln, Germany), and Cenovis (yeast extract from Switzerland, produced by the company Cenovis SA, Berstelstrasse, Switzerland) are other examples for commercial SCP products from yeast. Yeast extracts represent a good source of SCP and also five important vitamins of the group-B. *Torula* (*Candida utilis* that has been renamed as *Pichia jadinii*) is also commercially available as flavoring agent and is rich in protein. In the 1980s, the Provesta Corporation used *Torula* in their product Provesteen^®^T and used *Pichia* and *Kluyveromyces* yeast in other similar products [97]. *Torula* is used to replace the flavor enhancer monosodium glutamate because of its richness in glutamic acid.

Yeast spreads such as Marmite are produced from starch-derived glucose or from by-products of beer brewing, while the PEKILO process developed in Finland used lingo cellulosic sugars to produce SCP for animal feed. Moreover, yeast can grow on alkanes and methanol for SCP production. Methylotrophic yeasts as *Pichia pastoris* can produce biomass and protein from methanol. Phillips Petroleum Company produced *Pichia pastoris* at industrial scale and achieved yield of 130 g (DW)/l biomass and productivity of 10 g l^−1^ h^−1^ [98].

Another example of SCP from yeasts is the dairy substitute created from GM yeast; the animal-free ice cream launched by the company Perfect Day. Perfect Day’s animal-free ice cream is created by adding the DNA of cow milk to a yeast (GM yeast) to create dairy proteins (casein and whey) by fermentation. Those dairy proteins are afterwards combined with water and plant-based ingredients to create a dairy substitute that is used to make ice cream.

### 2.3. SCP from Filamentous Fungi

A wide range of filamentous fungi have been considered for use as SCP [13,52]. Products from *Fusarium* are commercially available. Fungal SCP generally contain protein content of 30–45% [13,52,99]. The amino acid composition of fungal SCP compares favorably with the FAO recommendations. The fungal SCP content of threonine and lysine is typically high, but methionine content is quite low, though still meeting the FAO/WHO recommendations [13]. SCP derived from fungi also provide vitamins primarily of the group-B. Additionally, the richness in glucans in fungal cell walls adds fiber to the diet. Fungi have a moderate nucleic acid content of 7–10%, which still is too high to be consumed by humans and thus requires processing to decrease it [13,100,101].

In 1985, Marlow Foods launched the Quorn brand, a meat substitute product made of mycoprotein from the filamentous fungus *F. venenatum*. This product is characterized by resemblance in texture between fungal biomass and meat products and has been broadly branded, marketed and sold for human nutrition. Quorn is the only SCP product solely used for human consumption. Quorn is generally regarded as a safe, well-tolerated food by regulatory bodies worldwide, including the FDA and the UK’s Food Standards Agency (FSA). Quorn is widely marketed in the United Kingdom, Belgium, the Netherlands, Sweden, Switzerland, and the United States. Since its introduction, it has become a well-established business with an annual sale of above USD 200 million [13,101]. Currently, there are a wide variety of Quorn-brand containing foods in the market such as Quorn burger, meatless chicken, and others. Freshly harvested mycoprotein has a protein content of 12% (*w*/*w*, wet weight) [101,102]. The protein digestibility is comparable to beef and soybean [13,103]. Moreover, it was reported that mycoprotein has a significant effect on appetite, especially satiety [104]. For that reason, mycoprotein is used for control of body weight and appetite, as well as for diabetes dietary [104]. It is generally known that some people are sensitive or allergic to an increased level of fungal material in their environment, thus adverse reactions to mycoprotein of Quorn products may occur with individuals with a history of mold allergies. From 1994 until 2000, the numbers of consumers raised from 2.25 to 13 million [103].

Another SCP product has been produced by the PEKILO process from the filamentous fungus *Paecilomyces varioti* grown on sugars in the sulphite waste liquor or wood hydrolysates. They had two factories, but as the cellulose mills ceased operations, the factories have been closed in 1991. Even though the product has been sold primarily as animal feed, it has been also studied as a supplement in meat products [13,29].

### 2.4. SCP Market 

According to Global Market Insights Inc., the size of SCP market in 2021 was above USD 8 billion and is expected to expand at a compound annual growth rate (CAGR) of more than 9% from 2022 to 2030, thus market size will exceed USD 18.5 billion by 2030 [105]. It is also anticipated that the SCP market size will exceed USD 4.5 billion by 2030 in EU countries. Advanced technologies’ integration in the food sector globally will adopt the growth of this industry. Development of innovative technologies for food processing will foster SCP industry trends in Europe. Market growth drivers include increasing adoption of production of dietary supplements, growing industrial production of livestock will motivate demand for animal feed protein, as well as growing number of malnourished populations worldwide. However, challenges that hinder market growth are SCP poor digestibility and having a high level of nucleic acids. The pandemic of COVID-19 elevated new challenges for the expansion of SCP market. The pandemic significantly disrupted the supply chains for the sector of food and feed worldwide. The food sector profoundly depends on microorganisms yet concerns of COVID-19 infection transmission via contact with these microorganisms resulted in a cessation in processing activities. Moreover, the restrictions related to the pandemic caused a temporary laboratories shutdown, giving more challenges for SCP industry [105].

North America and Western Europe are anticipated to significantly contribute to the growth of the SCP market over the forecast periods, since lifestyles are changing and demand for foods with added nutritional value is increasing in these regions. Asia Pacific and the Middle East are predicted to offer significant opportunities for SCP market growth over the forecast periods. The SCP market in the ASEAN region is led by Malaysia with value of USD 9.7 million in 2020 [105].

The size of the yeast market for animal feed application was estimated at above USD 1.5 billion in 2022 [105]. The increasing meat consumption worldwide will drive the industry to grow at an expected value of 7% CAGR from 2023 to 2032 (will reach USD 3.5 billion). Conspicuous R&D projects (e.g., constructing genetics labs, fermentation labs, and production halls or pilot plants equipped with advanced technology) will improve the benefits of yeast as animal feed, proposing massive development opportunities for market players, thus promoting the business outlook. Market growth drivers in North America are commitment to sustain quality of animal feed, in Europe they are robust application scope in the sector of livestock, and in Asia Pacific they are increasing meat domestic consumption and export. In Latin America, market growth drivers include rising beef exports, and in the Middle East and Africa the market growth driver is a strong poultry sector. However, challenges that hinder market growth are substitutes availability and spread of coronavirus. The market value of autolyzed yeast segment for feed is more than USD 500 million in 2022. The market value of yeast for poultry feed segment is expected to reach above USD 1.7 billion by 2032. The value of animal feed yeast market in North America in 2022 is USD 375 million at CAGR (2023–2032), more than 6.5%. Leading companies in the global animal feed yeast market include Cangzhou Tianyu Feed Additive Co., Ltd. (Chinese company that produces yeast powder as additives for feed), and Tangshan Top Bio-Technology Co., Ltd. (Chinese manufacturer of brewer’s yeast, autolyzed yeast, yeast extract, and natural, non-GM pure yeast powder as animal feed additive) [105].

According to Global Market Insights Inc., market size of yeast extract for human consumption is anticipated to reach more than USD 2.5 billion by 2030 [106]. Asia Pacific is planned to display greater grit in saving a leading position in the yeast-based spreads market. India is expected to significantly contribute to the universal growth of the market of yeast-based spreads over the forecast periods. European markets exhibit a rising in huge consumer base prevalence, and increasing local consumption of bakery products, which are major drivers for market growth in the region. Leading companies in the global yeast-based spreads market are Lallemand Inc. (Canadian company specialized in production of SCP from *S. cerevisiae* and Torula for human consumption), LeSaffre (French company that produces yeast (*S. cerevisiae*) and yeast derived products including SCP as well as yeast-based flavor ingredients), Bega Cheese (Australian company that produces Vegemite yeast-based spreads), and Unilever (producers of Marmite yeast-based spreads) [106].

In 2020, Quorn produced by the British company Marlow Foods Ltd. was the leading brand of meat substitute in Western Europe with a market share of about 16.7%, followed by the brand Linda McCartney produced by the British company Hain Frozen Foods UK Ltd. with a market share of about 3.9% [107]. However, Marlow Foods lately revealed that there is a 4.8% drop in revenues to 224.9 million pound in the year ended 31 December 2021 [108]. The company blamed the COVID-19 pandemic partly for this drop.

## 3. Edible Mushrooms

The mushroom is a large fleshy or woody fungus. Mushrooms are the higher fungi, belonging to the classes Ascomycetes (Morchella, Tuber, etc.) and Basidiomycetes (Agaricus, Auricularia, Tremella, etc.). They are characterized by having a heterotrophic mode of nutrition [109]. They are rich in protein and constitute a valuable source of supplementary food. The great value in promoting the cultivation of mushroom lies in their ability to grow on cheap substrates, industrial and agricultural waste (rich in nitrogen and carbon) [110].

The application of the term mushroom has been coined to edible species only (the term toadstool to those considered poisonous). Meanwhile, the species that result in death or serious illness when eaten are less in numbers. Mushrooms must be well investigated as in some cases even two closely related species, one of them might be poisonous while the second is edible. The great majority of mushrooms are tough, woody, bitter and tasteless. Fresh commercially grown mushrooms can always be eaten safely.

The mushroom species are usually grown commercially to attain a size of 5 to 10 cm tall and has a fleshy cap from about 2 to 10 cm across. Historically, mushrooms are cultivated commercially in caves, dark cellars, or specially constructed mushroom houses, in which the proper humidity and temperature are maintained. The commercial production of edible mushrooms occurs in buildings in which temperature and humidity are strictly regulated. A special bedding culture is prepared and inoculated with a pure culture of the fungus mycelium. Several crops of mushrooms are produced from each inoculation.

Edible mushrooms are variable and used as a standalone meal or in medicinal applications. Some mushrooms become more common and popular and can be found in local fresh-food grocery or in the hypermarkets. Popular and most commonly cultivated mushrooms in the world are belonging to the phylum Basidiomycota, in the genera Agaricus such as *Agaricus bisporus* (common button mushroom), Pleurotus such as oyster mushroom, Lentinus such as *Lentinula edodes* (shiitake), Auricularia such as *Auricularia ssp.* (wood ear or black ear mushrooms), Flammulina such as *Flammulina velutipes* (enoki or winter mushroom), Volvariella such as *Volvariella volvacea* (paddy straw mushroom), Tremella such as *Tremella fuciformis* (silver ear mushroom), Agrocybe such as *Agrocybe cylindracea*, nameko (*Pholiota nameko*), monkey head mushroom (*Hypsizygus marmoreus*), as well as two famous medicinal mushrooms, namely Maitake (*Grifola frondosa*) and the Reishi (*Ganoderma lucidum*) [111,112]. Some edible mushrooms are listed in Table 2. Commercial mushrooms are safer than those grow in the nature. Additionally, wild mushrooms are growing seasonally, in particular times and conditions. Meanwhile, some species are preferred to be collected from the wild.

Mushrooms add flavor, texture, and nutritional value to many dishes. Mushrooms are widely consumed throughout the world and are produced in a multi-stage process. China is the biggest producer of mushrooms and produces alone more than 75% of mushroom and truffles in the world. Japan comes second and the USA is the third largest producer, with productivity of 470,000 and 383,960 tons/year of mushrooms, respectively [19]. In western Europe, the most commonly consumed mushroom is *Agaricus bisporus*, but in the Far East, species of *Pleurotus*, and *Auricularia besides Lentinus edodes* are the dominant cultivated mushrooms [128].

Mushrooms are cultivated on various locally available substrates. Typically, they are produced in SSF on agricultural or farm wastes. Mushrooms are produced in most of the countries by individuals at home or on a large-scale by the companies. The presence of a suitable cheap substrate may favor the production of one type over the others. For example, *Agaricus bisporus* is grown in the UK, the USA, and France on wheat straw, *Volvariella volvacea* is grown in South-East Asia on damp rice straw and in Hong Kong on cotton waste, and *Lentinus edodes* is cultivated on fresh oak logs in Japan. In the industry, media for fungal fermentation need to be optimized regarding the specific application and production process [129].

## 4. Selected Fermented Foods of Fungal Origin

Fermented foods are produced nearly by every ethnic group. They vary based on food ingredients, microbes that induce the fermentation processes, the containers used in fermentation and storage, fermentation time and temperature, and such. Apparently, the main purpose of the fermentation processes is preserving foods and getting a new taste and aroma. However, in fact, the early understanding for the fermentation as a process has been broader. Foods have been fermented to get ethanol, which is one of the main chemical compounds used in mummification by the ancient Egyptians, due to its dehydration capability, antimicrobial activity, and volatility. At the same time, alcohol had been used as a solvent and as an extracting enhancer for the active ingredients of the medicinal plants. Fermentation produces stability in the foods that have been fermented, which means for early humans, a positive and unique approach. In their minds, the process that keeps food unspoiled should involve elements, factors, or even energy that will keep them healthy.

Modern science has a new explanation based on the experimentation and analysis of food ingredients. Thus, there are established facts about fermentation in today’s life, the most important of them is that fermentation increases the nutritional value of raw ingredients and that fermented foods improve our digestion. However, such early beliefs keeps many of fermented foods available until now. Many of them are consumed during religious celebrations and on particular days in the year. The old civilization believes in the combination of elements. For example, they believe in consuming the whole plant that contains active ingredients rather than extracting these active ingredients. Fermented foods gave early humans what they need; fermented foods have different ingredients in an unspoiled mixture and are available in the form of foods that are digestible, healthy, tasty, and cheap. Due to their highly positive properties, nearly every meal contained a dish made from fermented foods.

Due to the excessive modernization, such habits started to disappear slowly. However, this has led to discover, what such foods have given us, and why our grandfathers for generations keep them on their tables. Some of such foods have been treated with great care and kept as a secret for hundreds of years such as kefir. Some Caucasians believe that kefir has a miracle power, and if any foreigner discovers it, this power will disappear. It took an effort from the Kaiser of RUSSA to know the secret and to get a sample of it. This sample opened minds about the science of probiotics and kefir has been used for years as a treatment for every illness.

For some ethnic groups around the world, fermented foods are not just a tradition or a medicine but a matter of survival. Non-food material could be changed to be food only after being fermented. Nowadays, fermented foods—particularly yeasts and filamentous fungi-based fermented foods—can save humanity from the hunger.

Another important criterion of fermented foods is that some of the wild microbes with the ability to ferment foods can colonize our gastrointestinal system to digest food inside our intestine. Others cannot but can survive for days and also do their activities and help us in food digestion. For that and during hundreds of years, human blindly learns that some fermented foods should be eaten daily, weekly or even once a year.

Nowadays, some diseases have been prevented by the consumption of fermented foods. For example, the intake of fermented foods was reported to be a crucial step toward successful management of Alzheimer’s disease [130]. The explanation for this effect might be given to the increase in particular nutrients such as protein and vitamins. In contrast, unwanted compounds or those not good for our health such as the anti-nutritional chemicals including phytates, tannins and polyphenols might disappear due to the fermentation process or at least their amounts are reduced. For more details, refer to Kumar et al. (2022) and the references within [130].

The fermentation of food is often defined as the manufacture of foods employing the action of microorganisms and their enzymes. In Africa (e.g., Mozambique and Uganda), the use of mycelial fungi for fermentation is exemplified by their use in detoxification of bitter cassava roots [131]. There are a significant number of the fungal genera involved in fermented foods production. For more details, refer to [132,133,134] and the references within.

There are hundreds of beneficial yeasts, and the most famous is *S. cerevisiae*. It is used to produce bread, beer, and wine. *Saccharomyces* yeasts also form symbiotic relation with bacteria to form kefir [135]. Yeast can be found and isolated from different environments, particularly from the sugary mediums. There is a respective number of fermented foods that contain yeast [134,136].

The fungal fermented foods and products are common in Asia and Europe [15,137]. Filamentous fungi are used traditionally as starter cultures in Asia. They have several contributions, such as saccharification, and ethanol production. In Europe, they are used in developing different dairy products particularly in the ripening processes of various types of cheese (e.g., Roquefort, Camembert) and for enzyme production [134,138,139]. Within the genus Aspergillus, there are important species such as *Aspergillus acidus*, *A. oryzae*, *A. niger*, *A. sojae*, *A. sydowii*, *A. versicolor* and *A. flavus*, which are used in traditional fermented foods production such as soya sauce fermentations, Miso, sake, awamori liquors, and Puerh tea [137,140,141,142].

### 4.1. Kefir

Kefir is fermented by both of bacteria and yeast. It is a fermented milk drink (cow, goat, or sheep milk with kefir grains) similar to a thin yogurt or ayran [143,144,145,146,147,148,149,150,151,152]. It had been coined in Caucasus in the 1900s. The grains are composed of colonies of living Lactobacilli and yeast. They live together in a symbiotic relationship and ferment milk at room temperature. Traditional kefir has been made in goatskin bags that have been hung near a doorway. The bag has been knocked by anyone passing by to keep milk and kefir grains well mixed. Such a practice is usually also done by traveler Bedouin in the Middle East. The kefir grains are initially created by auto-aggregations of *Lactobacillus kefiranofaciens* and *Saccharomyces turicensis*. They are biofilm producers. In addition, they can adhere multiply to the surface to become a three-dimensional micro-colony [153,154]. Yeasts found in kefir include *Candida kefir*, *S. cerevisiae*. *Kluyveromyces marxianus*, *Kluyveromyces lactis*, *Saccharomyces fragilis*, *Torulaspora delbrueckii*, and *Kazachstania unispora* [155]. During the fermentation process, the lactose is digested, which makes kefir ideal for those persons that are with lactose intolerance. Kefir possesses natural antibiotics and rich in the B1 and B12 vitamins, calcium, folic acid, phosphor, and K vitamin.

### 4.2. Tempeh

Tempeh has been firstly produced in Indonesia for thousands of years. It remains the most important staple food there and an inexpensive source of dietary protein. It is being spread to other countries such as Malaysia and the Netherlands. Tempeh is made from partially cooked fermented soybeans. It is especially popular on the island of Java. Indonesian tempeh, of Javanese, is in form of soybean cakes. It is prepared by making a natural culture and process of controlled fermentation. *Rhizopus oligosporus* is used in this fermentation process (known as tempeh starters) [156,157]. The main step to make tempeh is the fermentation of soybeans. *Rhizopus ssp*. is used to inoculate the soybeans. Traditionally, a previously fermented tempeh is mixed. It contains the spores of *Rhizopus oligosporus* or *Rhizopus oryzae*. The mixture then spread to a form of a thin layer and allowed to ferment for 24 h at a temperature around 30 °C. The soybeans must get cold to allow the germination of the spore. Typically, tempeh is ripened after 48 h. The fermented product has distinguishable whitish color, firm texture, and nutty flavor. Fermented soybeans in tempeh contain B12 vitamin, a byproduct of the fermentation process. Tempeh enhances the body’s absorption of the isoflavones. The fermentation process also removes the enzyme inhibitors that occur naturally in soybeans. Tempeh is high in fiber, beneficial bacteria, enzymes, and manganese. It is a good digestible protein source and a great meat substitute (high in protein). It is also a good source of calcium and contains all essential amino acids. Sometimes, it is made from a blend of grains, beans, or other vegetables.

### 4.3. Miso

A product of fermented soybeans, which is produced and consumed in the Far East. Miso came during an early time in China and Korea. However, Japan nowadays is the main producer and consumer. It is estimated that 5k/person/year are consumed in Japan [158]. In traditional practice, miso is prepared using a seed culture of the other previously produced miso. A seed culture usually contains some yeast and bacteria. The most common yeast is *Z. vousir* and *C. versetilis.* Miso is also prepared with salt and koji (mold *Aspergillus oryzae*). Sometimes, rice, barley, algae, or other ingredients are added. In Japan, miso of fish had been manufactured since the Neolithic time [159]. Miso is rich in protein, vitamins, and minerals. It can be produced as very salty or very sugary. Miso fleshy fermented dough is made by grinding some soybeans, malted rice or barley and salt together, then used specially to make soups and sauces. The natural miso is a living food that contains many beneficial microorganisms [160]. *Tetragenococcus halophilus* is an example. Those microbes can be killed by overcooking.

## 5. Safety of Fungal Proteins

Fungal proteins as any other product used for food or feed need to be safe to produce and use. Regulations exist in most regions to ensure that human food or animal feed are safe for consumption and these regulations differ depending on whether products are expected to be food (providing nutrition along with pleasant taste and aroma) or will be marketed as additives for food (texture modifiers, preservatives, etc.) or applied as feed and additives for feed [161]. The pleasant taste and aroma of the products made of fungal proteins will help in raising demand for them and boosting the appeal of these novel protein sources in a crowded marketplace. As discussed above, three crude protein products from fermentation (by-) products from yeast and filamentous fungi are approved as feed by Commission Regulation (EU) No 68/2013. Additionally, three SCP fungal strains are accepted for food use in EU countries, only one of them (the yeast *Yarrowia lipolytica*) has been authorized via the Novel Food Regulation ((EU) 2017/2470), and its use is restricted to food supplements. Molitorisová et al. (2021) mapped the regulatory environment that governs mushrooms and mycelium products (MMP) in EU countries—food law provisions applicable to MMP produced or marketed in the EU [162]. They found that the sector is still in the developing phase, and regulatory framework application to MMP comprises numerous legal doubts. The law classifies MMP as foods or medicines based on the proposed use. Novel MMP could be classified as medicines. This classification can exclude provisions of food law. Operators of food business that work with borderline products (food/medicine) should consider their claims. Mushrooms and mycelia, along with products derived thereof, can be subjected to the common agricultural policy rules. The classification of MMP as novel foods is challenging. As an example, regardless of a long history of some fruiting bodies consumption, products derived from mycelium of the same species may be subjected to novel foods regulation ((EU) 2015/2283). This is similarly the case of species that are not consumed commonly. The Novel Food Regulation obligates important regulatory requirements on applicants, who applying for authorizations of novel food. These regulatory requirements include safety proof via robust and solid scientific evidence, such as several studies of toxicity, animal models, or human data. MMP are generally classified as “vegetarian” or “vegan” in EU countries and bear claims of sustainability.

The name of the raw materials or substrates used in SCP production represents the main safety hazard. For example, the possible presence of carcinogenic hydrocarbons in n-paraffin or gas oil or presence of heavy-metal contamination in the mineral salts and solvents after extraction. Quorn is produced in a chemically defined medium from glucose (hydrolyzed starch) in a well-defined process which meets international standards [28]. It is a privilege to use a standard medium, since the fungal nutritional composition differs based on the changes in the medium composition [119]. Meanwhile, the economic production of fungal proteins in some cases (e.g., the production of edible mushrooms) favors the use of agriculture wastes. Nevertheless, such wastes should be free from any toxic or harmful compounds (e.g., pesticides).

Dried and heat-killed yeast biomass used as a protein source must be safe for both human and animal nutrition in accordance with the current food and feed safety regulation. Additionally, the concentration of heavy metals in yeast biomass should be low and must not exceed the EU threshold values [163]. In certain cultivation and growth conditions, fungi could produce specific secondary metabolites. For that reason, product quality assurance, good manufacturing practice (GMP) as well as monitoring fungal molecular properties should be performed to ensure that fungal products are consistently produced and controlled according to quality standards [164]. In 2018, King et al. performed genomic analysis (using shotgun sequencing) to differentiate between *F. venenatum* (Quorn fungus) and *F. graminearum* (closely related phytopathogen), comprising genes that code for different mycotoxin types (type A trichothecene mycotoxin TRI5 cluster in case of *F. venenatum* and type B trichothecene mycotoxin TRI5 cluster in case of *F. graminearum*) [165]. They identified differences between the genomes of the two fungal species that could participate in their contrasting lifestyles and highlighted *F. graminearum*-specific candidate genes potentially required for pathogenesis. So far, Quorn fungus is not known to produce mycotoxins in the used processing conditions, even though regular monitoring is still performed to ensure that the final product contains none of these toxins [166].

It is wise to simplify the production methods for certain fungal species (e.g., mushrooms) to be in-house, reliable, and applicable for the farmers. Even so, the end product quality and safety should be investigated [167]. In addition, the use of waste-derived substrates for SCP production needs public acceptance of waste-derived foods in addition to the safety regulation. The public acceptance can help countries to set their own safety standards, nevertheless, those standards should be science-based [168]. Products with a short shelf life, particularly products from viable fungi such as fresh mushrooms, which are prone to spoilage should be continuously checked and properly stored in the markets.

Vital safety concerns of SCP include microbial toxins either from SCP producing microorganisms or contaminants, SCP RNA content, potential allergy reactions, and harmful substances derived from production raw materials. Srividya et al. (2013) and Ukaegbu-Obi, (2016) highlighted that the suitability of SCP as a source of edible protein should be considered individually because, such as any food, it might cause different allergic reactions based on the individual sensitivity to the fungal protein [31,36]. In addition, the inactivation or killing of the viable fungal cells is so important. EFSA (2008) reported that subjects who are exposed to viable yeast inhalation are particularly at health risk [169].

The key concern of food safety associated with mycoprotein is allergens [170]. Data are limited on this aspect, but adverse effects to mycoprotein consumption were reported in patients with a history of allergies to molds. Type I hypersensitivity reactions were found in a 27-year-old mold allergic female patient within a few minutes of consuming Quorn burger [171]. Likewise, Sandhu and Hopp (2009) reported that 15-year-old male patient with a history of allergies to several molds showed type I hypersensitivity reaction to ingestion of meatless chicken (Quorn) [172]. In 2018, Jacobson and DePorter analyzed self-reported adverse reactions related to Quorn-brand containing foods from 1752 individuals and found that most of these reactions involved allergies such as anaphylaxis and hives or gastrointestinal symptoms as diarrhea and vomiting [173].

Yeast is most appropriate for SCP production due to its superior nutritional quality. Cereals supplemented with SCP derived from yeast have been proved to be as good as animal proteins [174]. Yeast for use as SCP is characterized by the absence of toxic and carcinogenic compounds biosynthesized by yeast from the substrates or formed during processing. In addition, about 100 pounds of yeast will produce 250 tons of proteins per day. However, the use of yeast for human and animal consumption may be limited by a high nucleic acid content, which is mostly metabolized to yield uric acid possibly at high levels leading to renal stones, and low cell-wall digestibility [175].

Cautiously choosing SCP producing organisms (e.g., fungi that produce mycotoxins cannot be chosen as they may cause allergic reactions, carcinogenesis, or even death) [176], the process conditions, and the product formulation will overcome toxins challenge. SCP RNA content could be decreased to acceptable levels. The consumption of yeast protein with high DNA content might cause gut and kidney stone for those who have purine metabolism malfunction [52,177]. Only 1% for short time are recommended in feed or food [52]. SCP product that contains DNA higher than 1% is allowed only as a feed to short life-span animals [178].

The safety concerns of fungal proteins including SCP could be summarized in the following points:A need to specify the unique properties for fungus to be claimed/and used as a protein producer or as SCP;Using internationally standard fungal strains in protein production or as SCP and suitable culture conditions for maintaining fungi;Good manufacturing practice rules must be followed and controlled by the quality control lab (QC) and quality assurance lab (QA), labeling (e.g., avoidance instruction for some health conditions), as well as defining minimum/maximum consumed amount/day. Other effectors must be controlled such as the shelf life conditions;The accepted amount that could be consumed concerning sex/age/weight and such;Consumer feedback concerning any adverse effects or health problems;It is preferred to use chemically defined media as well as a well-defined process that meets international standards to achieve consistent production of fungal proteins; Globalization of GMP, QC, QA, market practices, feedback, consumer service, etc. is necessary;Restricted governmental laws for safety regulation should be available;Inactivation of the used fungi in the end product is a crucial step and must be applied;In case of waste usage (e.g., agriculture wastes in mushrooms cultivation), they should be free from any toxic components or heavy metals (and such) to guarantee safe final products;The effect of cultivation conditions (other than media) on the microbes should be under control to avoid the production of any undesired secondary metabolites;The nucleic acid content should not exceed 1%. In addition, the product that contains more than 1% nucleic acids should be fed to short life-span animals;Products with short shelf life, particularly products from viable fungi such as fresh mushrooms, which are prone to spoilage should be continuously checked and properly stored in the markets;Mycotoxins are known fungal products. Some safe fungal production processes under certain changes in the production conditions might lead to production of mycotoxins or other types of toxins. Only experts should decide if the proposed changes in the production conditions are accepted or not, therefore the new changes should be under investigations concerning the end product quality. One should not neglect any physical, chemical or biological measures;The production process is not the end point. The end point will be when consumers utilize fungal products safely and for a long time. For that, investigating the effect of products on consumers (either human or animals) should not stop;GM fungi products should be labeled and should be deactivated from any genetic elements.

## 6. Conclusions

SCP producing fungi grow as single or filamentous individuals rather than as complex multicellular organisms such as plants or animals. Use of fungal biomass as a protein source gives many advantages over the conventional sources. Microbes including fungi have a shorter generation time, utilize many substrates, do not need arable land or any particular season to grow, and can be produced continuously in any part of the world. The product yield varies according to the substrate and type of fungi.

## Figures and Tables

**Table 1 jof-09-00073-t001:** Examples of different fungal strains that utilize different types of waste as substrate for SCP production and percent of their protein composition.

Waste	Fungal Strains	Protein (%)	References
Apple pomace	*Aspergillus niger*	17–20	[62]
Apple peels	*Trichoderma harzianum*	21.65	[63]
Banana peels	*Trichoderma harzianum* *Rhizopus oryzae* *Saccharomyces cerevisiae*	17.60 15.37 9.99	[57,63]
Spent grain from Brewery	*Rhizopus oligosporus*	32.90	[64]
Cactus pear	*Aspergillus niger* *Rhizopus ssp.*	5.2	[46]
Tomato waste	*Trichoderma harzianum*	84.46	[63]
Mango waste	*Trichoderma* *harzianum*	33.38	[63]
Shell wastes of Prawn	Unspecified marine yeast	61–70	[65]
Wheat straw	*Pleurotus ostreatus var florida*	63	[66]
Orange pulp, molasses, Spent grain from Brewery, whey, pulp of potato	*Kluyveromyces marxianus*	59	[67]
Soy molasses	*Candida tropicalis*	56	[68]
Inulin, crude oil, glycerol waste hydrocarbons	*Yarrowia lipolytica*	48–54	[69,70]
Waste liquor	*Aspergillus niger*	50	[71]
Stick water	*Aspergillus niger*	49	[72]
Spoiled fruits of date palm	*Hanseniaspora uvarum*	49	[73]
Low quality fruits of some dates	*Saccharomyces cerevisiae* ATCC64712	55	[74]
Cucumber and orange peels (leftovers)	*Saccharomyces cerevisiae*	Cucumber peel (53.4) Orange peel (30.5)	[55]
Candies production effluent and agricultural digestate	*Saccharomyces cerevisiae*	28	[75]
Banana peel, carrot pomace, citrus peel, and potato peel	*Saccharomyces cerevisiae*	47.7	[76]
Fish, citrus peels, banana, pineapple, and apple	*Saccharomyces cerevisiae*	40.2	[77]
Oat bran, rye bran, and rye straw hydrolysates	*Yarrowia lipolytica*	30.5–44.5	[78]
Cheese whey	*Kefir* ssp.	54	[79]
Cheese whey	*Kluyveromyces marxianus*	43	[60]
Cheese whey filtrate	*Trichoderma harzianum*	34	[80]
Cheese whey	*Kluyveromyces marxianus Candida krusei*	48	[81]
Poultry litter, waste powder of capsicum	*Candida utilis*	Poultry litter (29) Waste powder of capsicum (48)	[82]
Waste from processing of potato starch	*Aspergillus niger* H3 (mutant strain)	38	[54]
Glucose	*Fusarium venenatum*	44	[83]
Lignin	*Chrysonilia sitophila*	39	[84]
Citrus pulp	*Trichoderma virideae*	32	[85]
Spent grains from Brewery, hemicellulosic hydrolysate	*Debaryomyces hansenii*	32	[85]
Bagasse	*Candida tropicalis*	31	[86]
Orange pulp, molasses, spent grains from Brewery, whey, potato pulp	*Saccharomyces cerevisiae*	24	[67]
Banana wastes	*Aspergillus niger*	18	[87]
Rice bran	*Aspergillus niger*	11	[88]
	*Aspergillus flavus*	10	
	*Fusarium semitectum*		
	*Cladosporium cladosporioides*		
	*Penicillium citrinum*		
	*Aspergillus ochraceus*		
Rice bran (deoiled)	*Aspergillus oryzae*	24	[89]

**Table 2 jof-09-00073-t002:** Some edible mushrooms and their characteristics.

Mushroom Commercial Name	Mushroom Scientific Name	Characteristics	References
White bottom mushroom	*Agaricus bisporus*	- Native to north American and Eurasian grasslands. - First grown domestically in the 1650s, and first scientifically described in 1707. - Found in fields and grasslands following rain. - It contains 3.09 g protein/100 g mushroom.	[113]
Shiitake mushrooms	*Lentinula edodes*	- Grow predominantly in China and Japan. - Have woody taste. - Popular in Asia and used to prepare many local dishes.	[114]
The chanterelle	Several species in the genera *Cantharellus, Craterellus, Gomphus,* and *Polyozellus.*	- Have nut-like flavor. - Popular in Europe since ancient Roman times. - Abundant in midsummer in coniferous and hardwood forests. - Grow from 5 to 10 cm tall. - Have an orange or yellow cap that is funnel shaped when young. - Crisp and heavy species are favored for eating.	[115]
The edible pore mushrooms	*Boletus*	- Grow during summer and early autumn in open deciduous woods. - The king mushroom has a stem of 5–15 cm tall and a brown cap of 10–15 cm across. - These mushrooms are most tender when the veins that cover their cap are pale yellow. - Mostly ectomycorrhizal fungi.	[116]
Oyster mushroom	*Pleurotus ostreatus*	- Has a pleasant, oyster-like flavor. - Often dipped in egg and fried slowly. - Grows on decaying tree trunks in bracket-like clusters. - It is almost stemless. - Abundant from June to November.	[117,118,119,120,121]
Portobello mushroom	*Agaricus bisporus*	- *Agaricus bisporus* marketed in its mature state (brown colored with a cap of 10 to 15 cm) is called Portobello. - Has meaty texture and strong earthy flavors.	
Enoki mushrooms	*Flammulina velutipes*	- Very popular in many Asian dishes. - Have a mild earthy flavor. - Have crispy texture when served in soups.	[122]
Jelly Ears	*Auricularia auricula-judae*	- Very common in Asian countries. - In China, it is also used in medicine.	[123]
Porcini mushroom	*Boletus edulis*	- Described as a bit nutty and meaty. - It has porcini smelling such as freshly baked.	[124]
The sulfur mushroom	*Laetiporus sulphureus*	- Develops on rotten logs, stumps, standing trees, producing a brown wood rot. - They may reach a breadth of several meters and a weight of several kilograms. - Spores are produced in enormous numbers in minute pores on the lower surface. - The fungus is edible if picked in the young, growing stage, yet rapidly becomes dry, tough, and honeycombed by insect larvae.	[125]
The morel mushroom	*Morchella esculenta*	- Has unique irregular honeycombed appearance and excellent flavor. - Morels are found during the spring in old orchards especially in areas that have been burned out. - Most highly prized of all mushrooms.	[126]
The shaggy-mane	*Coprinus comatus*	- The shaggy-mane is a common and widespread mushroom species appearing from spring until fall in lawns, gardens, and other open spaces. - The shaggy-mane is considered one of the most sought after edible species. - Blackened portions should be discarded before the mushroom is eaten. - Some related species cause poisoning when alcohol is consumed within five days after ingestion of the mushroom.	[127]
The giant puffballs	*Calvatia gigantea*	- They are very large and globose and have no gills or pores - Their spores are born internally. - Their fruiting body is creamy white in the edible stage. - Grow during late summer in grassy places and at the edge of woods. - They significantly differ from poisonous or offensive fungi. - They are edible as long as the tissues within are not discolored or larva-infested. Species that are brown to purple within, should be avoided.	

## Data Availability

Data is contained within the article.

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
