# Peer review of "Fungi as a Source of Edible Proteins and Animal Feed"

_jof, 2023, doi:10.3390/jof9010073_

Round 1

Reviewer 1 Report

The manuscript is a review that presents an overview over single cell protein production and use thereof for edible food. It aimed to address single cell protein production from yeast and filamentous fungi, mushrooms and fermented food products, as well as safety and consumption concerns. The contribution is insightful and comprehensive, and will serve to a wide readership from across various fields in the context of sustainable food systems. The manuscript is clear, relevant for sustainable food systems and anyone involved in various fields involved, and presented in a well-structured manner 

While the review is comprehensive and insightful, the manuscripts lacks statements regarding the relevance of the review and the knowledge gap that the manuscript seeks to fill. In addition, aim and objectives of the review should be more concisely articulated.

Of concern is that the title of the manuscript does not convey that the review is also about single cell protein in the context of animal feed. As it stands, the reader may expect an article solely in the context of human food. 

When reading the text, it is not always clear whether all fungi and substrates that are mentioned are safe for human consumption or in the use of feed for animals. For example, the EU’s regulations on novel food are not mentioned nor are GRAS (generally regarded as safe) species and applications. Largely, an overview of single cell protein production is presented but there are instances where statements are lacking regarding how mentioned aspects link back to use in human food or animal feed. 

Another concern that needs to be addressed is the conceptualization of ‘waste’ in this manuscript. Food made from waste for human consumption is not legal in various places across the world. Yet, food can be made from recovered edible/safe to consume sources that may otherwise not find their way back to human consumption in a circular manner such as leftovers, biproducts, surplus products and residues. The conceptualization regrading this needs to considered by the authors. 

Where references were cited, these are appropriate but most have not been published within the past five years. Yet, for a review article, the manuscript lacks references in various paragraphs throughout the text. Please see further comments referring line numbers below. 

The conclusions are consistent with the evidence and arguments presented, and the ethics statements and data availability statements are adequate.

There are some specific comments which I hope will help improve the manuscript:  

The link to use in human food or animal feed needs to be made consistently throughout the article. It is missing in several instances, examples of which include aspects mentioned in line 208-214; and 220-221.

The cited references are mostly not recent publications (not within the last 5 years; the oldest from the 60s). Perhaps more recent references could be cited in at least some instances.

Citations should be amended appropriately in the following sections: lines 31-46; 61-76; 107; 121-125; 174-191; 208-214; 220-221; 250-255; 257-260; 266-268; 271-274; 293-294; 301-302; 311-312; 342-376; 379-385; 389-427; 430-434; 442-443; 448-449; 459-465; 469-471; 473-478; 479-491; 493-505; 531-537; 547; and 549-551. In line 300, the named company’s full details may be amended in the same way as was done for the yeast producing companies (e.g., line 292). In addition, the sentence in line 323-325 is refers to ‘several’ studies but only one citation is presented.

Table 1 omits relevant substrates. While the table is easy to understand, it misses recently published studies that focused on using leftovers, surplus- and biproducts and residuals that may otherwise have been lost in the production of single cell proteins (e.g., bread, buckwheat, oat, dates etc.)

Some minor comments: 

Please make sure to introduce abbreviations such as ‘SCP’ where appropriate (e.g., line 61).

For readability, the second last paragraph of the introduction (lines 81-98) could be split into several paragraphs with their own focus in their own right. I suggest one about patents (including aspects mentioned in lines 91-93);  one about the aspects mentioned in lines 87-91; and one about ‘mycoprotein’ and its characteristics. 

In terms of subheading 2.1.3. (Fermentation process), it may be amended that the requirements mentioned in lines 239-242 are specific to the context of submerged fermentation (SmF).

Please amend what the abbreviation ‘RNA’ in line 276 stands for.

In line 286-287, the use of SSF over SmF in the production of animal feed is mentioned. It may be noted that a review paper in the context of single cell protein for edible food should also mention and refer to recent studies in which SSF was used over SmF in the production of human food products.  

In subsection 4.2. (line 472) the authors may choose either one of the two ways of spelling this food product as long as they ensure that the same spelling is then used throughout the manuscript. 

In line  589, it is unclear what the authors mean by ‘unsold products in the market’.

Author Response

Dear Editors and Reviewers,

We are submitting here the revised version of our manuscript entitled “Fungi as a source of edible proteins and animal feed” by Amro A. Amara and Nawal Abd El-Baky (manuscript ID: jof-2111473) for potential publication in your respected journal Journal of Fungi.

We are thankful to the reviewers for their useful remarks and critiques. We address all the issues raised by the reviewers and revised manuscript accordingly. Please find below our point-by-point response to reviewers’ critiques.

We sincerely believe that the suggestions improved the quality of our work and hope that the manuscript became more suitable for publication in Journal of Fungi.

Sincerely,

Amro A. Amara and Nawal Abd El-Baky

Reviewer 1

Please note that Text highlighted in yellow, represent places where new citations were added.

Comment 1: While the review is comprehensive and insightful, the manuscripts lacks statements regarding the relevance of the review and the knowledge gap that the manuscript seeks to fill. In addition, aim and objectives of the review should be more concisely articulated.

Response: Abstract and introduction were modified accordingly. 

Comment 2: Of concern is that the title of the manuscript does not convey that the review is also about single cell protein in the context of animal feed. As it stands, the reader may expect an article solely in the context of human food.

Response: Title was modified accordingly.   

Comment 3: When reading the text, it is not always clear whether all fungi and substrates that are mentioned are safe for human consumption or in the use of feed for animals. For example, the EU’s regulations on novel food are not mentioned nor are GRAS (generally regarded as safe) species and applications. Largely, an overview of single cell protein production is presented but there are instances where statements are lacking regarding how mentioned aspects link back to use in human food or animal feed. 

Response: The EU’s regulations on novel food and GRAS (generally regarded as safe) species and applications were added as suggested. Additionally, for each case, intended application as food or feed was mentioned if available.

Comment 4: Food can be made from recovered edible/safe to consume sources that may otherwise not find their way back to human consumption in a circular manner such as leftovers, byproducts, surplus products and residues. The conceptualization regrading this needs to considered by the authors.

Response: This conceptualization was detailed in the manuscript as suggested.

Comment 5: Where references were cited, these are appropriate but most have not been published within the past five years. Yet, for a review article, the manuscript lacks references in various paragraphs throughout the text. Please see further comments referring line numbers below.

Response: Missed citations were added throughout the text. Also, the reference list was updated with recent citations in the last five years as recommended. 

Comment 6: The link to use in human food or animal feed needs to be made consistently throughout the article. It is missing in several instances, examples of which include aspects mentioned in line 208-214; and 220-221.

Response: In the mentioned two cases, the intention was biomass production or waste treatment, yet, for each other case, intended application as food or feed was mentioned if available.  

Comment 7: Citations should be amended appropriately in the following sections: lines 31-46; 61-76; 107; 121-125; 174-191; 208-214; 220-221; 250-255; 257-260; 266-268; 271-274; 293-294; 301-302; 311-312; 342-376; 379-385; 389-427; 430-434; 442-443; 448-449; 459-465; 469-471; 473-478; 479-491; 493-505; 531-537; 547; and 549-551.

Response: Missed citations were added throughout the text. Also, the reference list was updated with recent citations in the last five years as recommended.

Comment 8: In line 300, the named company’s full details may be amended in the same way as was done for the yeast producing companies (e.g., line 292). In addition, the sentence in line 323-325 is refers to ‘several’ studies but only one citation is presented.

Response: Corrections were done as suggested.

Comment 9: Table 1 omits relevant substrates. While the table is easy to understand, it misses recently published studies that focused on using leftovers, surplus- and biproducts and residuals that may otherwise have been lost in the production of single cell proteins (e.g., bread, buckwheat, oat, dates etc.)

Response: oat, dates, leftovers, etc. were added as suggested. But in case of   buckwheat, was used to produce l-carnitine enriched oyster mushroom (reference 118 in the manuscript).

Some minor comments: 

Comment 1: Please make sure to introduce abbreviations such as ‘SCP’ where appropriate (e.g., line 61).

Response: Done as suggested.

Comment 2: For readability, the second last paragraph of the introduction (lines 81-98) could be split into several paragraphs with their own focus in their own right. I suggest one about patents (including aspects mentioned in lines 91-93);  one about the aspects mentioned in lines 87-91; and one about ‘mycoprotein’ and its characteristics. 

Response: Done as suggested.

Comment 3: In terms of subheading 2.1.3. (Fermentation process), it may be amended that the requirements mentioned in lines 239-242 are specific to the context of submerged fermentation (SmF).

Response: Done as suggested.

Comment 4: Please amend what the abbreviation ‘RNA’ in line 276 stands for.

Response: Done as suggested.

Comment 5: In line 286-287, the use of SSF over SmF in the production of animal feed is mentioned. It may be noted that a review paper in the context of single cell protein for edible food should also mention and refer to recent studies in which SSF was used over SmF in the production of human food products.  

Response: Done as suggested.

Comment 6: In subsection 4.2. (line 472) the authors may choose either one of the two ways of spelling this food product as long as they ensure that the same spelling is then used throughout the manuscript. 

Response: Done as suggested.

Comment 7: In line 589, it is unclear what the authors mean by ‘unsold products in the market’.

Response: The sentence was rewritten.

Reviewer 2 Report

The paper is well written and the subject is interesting but consider doing a deep dive over the subject and presenting some innovation in this subject. There are some considerations below:

- add some market information such as the main type of products,  SCP financial value, and market growth tendencies.

- Market information could be crossed with the respective bioprocess according to each group of microorganism (e.g. solid-state fermentation, advantages to use wastes as substrate for SCP production) 

the differences between bioprocesses for mainly fungi products

Author Response

Dear Editors and Reviewers,

We are submitting here the revised version of our manuscript entitled “Fungi as a source of edible proteins and animal feed” by Amro A. Amara and Nawal Abd El-Baky (manuscript ID: jof-2111473) for potential publication in your respected journal Journal of Fungi.

We are thankful to the reviewers for their useful remarks and critiques. We address all the issues raised by the reviewers and revised manuscript accordingly. Please find below our point-by-point response to reviewers’ critiques.

We sincerely believe that the suggestions improved the quality of our work and hope that the manuscript became more suitable for publication in Journal of Fungi.

Sincerely,

Amro A. Amara and Nawal Abd El-Baky

Reviewer 2

Comment 1: Add some market information such as the main type of products, SCP financial value, and market growth tendencies.

Response:  Recent market data were added as suggested.

Comment 2: Market information could be crossed with the respective bioprocess according to each group of microorganism (e.g. solid-state fermentation, advantages to use wastes as substrate for SCP production) the differences between bioprocesses for mainly fungi products.

Response: Data on costs of substrates and fermentation were added in the appropriate positions in the text as suggested.

Reviewer 3 Report

General Comments:

In this study, it mainly focuses on fungal-derived proteins, fungal fermented foods, and some safety standards. The main problems include that the connection between protein in grain and protein in fungi is not clear, the literature is less cited, and the expression lacks the author 's own point of view. When it comes to security issues, it does not mention the potential insecurity factors of SCP, but only introduces some standards. Therefore, this submission is not recommended for publication in its current form.

Specific Comments:

1. Abstract: Lack of evidence linking food resources to protein sources.

2. Introduction: Fewer references cited and lack of some evidence.

3. Introduction: The second section provides solutions due to the pressure of agricultural production. This paragraph has little to do with the article and is not logical.

4. Introduction: The previous content did not mention security issues, the seventh paragraph appears a little abrupt.

5. Sections 2: Both Spirulina and Algae (seaweed) belong to plants, not fungi.

6. Sections 2.1: Fewer references cited and lack of some evidence.

7. Sections 2.1.2: Lignocellulosic waste not only can be used for mushroom production.

8. Sections 4.2: The meaning expressed by “Tempe” is not clear and is not mentioned in the text.

9. Tables:

- The format of Table 2 is problematic and needs to be modified.

10. Please check the punctuation carefully and then modify it.

Author Response

Dear Editors and Reviewers,

We are submitting here the revised version of our manuscript entitled “Fungi as a source of edible proteins and animal feed” by Amro A. Amara and Nawal Abd El-Baky (manuscript ID: jof-2111473) for potential publication in your respected journal Journal of Fungi.

We are thankful to the reviewers for their useful remarks and critiques. We address all the issues raised by the reviewers and revised manuscript accordingly. Please find below our point-by-point response to reviewers’ critiques.

We sincerely believe that the suggestions improved the quality of our work and hope that the manuscript became more suitable for publication in Journal of Fungi.

Sincerely,

Amro A. Amara and Nawal Abd El-Baky

Reviewer 3

Please note that Text highlighted in yellow, represent places where new citations were added.

General comment: In this study, it mainly focuses on fungal-derived proteins, fungal fermented foods, and some safety standards. The main problems include that the connection between protein in grain and protein in fungi is not clear, the literature is less cited, and the expression lacks the author's own point of view. When it comes to security issues, it does not mention the potential insecurity factors of SCP, but only introduces some standards. Therefore, this submission is not recommended for publication in its current form.

Response: The abstract and introduction sections were rewritten, to clearly address the aim of the work, the competitiveness between plant or animal proteins and fungal proteins as alternative source for edible protein and animal feed, potential insecurity factors of SCP, and missed citations were added.   

Specific Comments:

Comment 1. Abstract: Lack of evidence linking food resources to protein sources.

Response: The abstract was modified accordingly.

Comment 2. Introduction: Fewer references cited and lack of some evidence.

Response: Missed citations were added and introduction section was rewritten

Comment 3. Introduction: The second section provides solutions due to the pressure of agricultural production. This paragraph has little to do with the article and is not logical.

Response: The best competitor to fungal or in general microbial proteins is plant proteins for vegeterians or for those seek healthy and cost-effective foods, thus we should present the problem with plant and animal-derived proteins and their shortage to introduce for fungal protein.

Comment 4. Introduction: The previous content did not mention security issues, the seventh paragraph appears a little abrupt.

Response: The introduction was modified accordingly.

Comment 5. Sections 2: Both Spirulina and Algae (seaweed) belong to plants, not fungi.

Response: Both were deleted as suggested.

Comment 6. Sections 2.1: Fewer references cited and lack of some evidence.

Response: rewritten and citations were added.

Comment 7. Sections 2.1.2: Lignocellulosic waste not only can be used for mushroom production.

Response: the sentence was deleted.

Comment 8. Sections 4.2: The meaning expressed by “Tempe” is not clear and is not mentioned in the text.

Response: it is the other name of Indonesean fermented food temph and description is in the text.    

Comment 9. Tables: The format of Table 2 is problematic and needs to be modified.

Response: Done as suggested.

Comment 10. Please check the punctuation carefully and then modify it.

Response: Done as suggested.

Round 2

Reviewer 1 Report

The authors did a great job at improving the manuscript.

Please find a few comments that arose while reading the revised version.

line 120-124: Please be very careful to make sure the statement in the sentence regarding Novel Food in the EU is correct. As to the latter part of  the sentence, some filamentous fungi are considered novel (e.g., Neuropsora intermedia which is commonly used in Indonesia to make omcon). 

line 125-133: there are other fungal strains that are not considered novel but it is good that three examples are provided.

line 491: please move the references from the subheading and instead cite references in the text.